# CustomNet: Object Customization with Variable-Viewpoints in Text-to-Image Diffusion Models

Submission Id: 3741

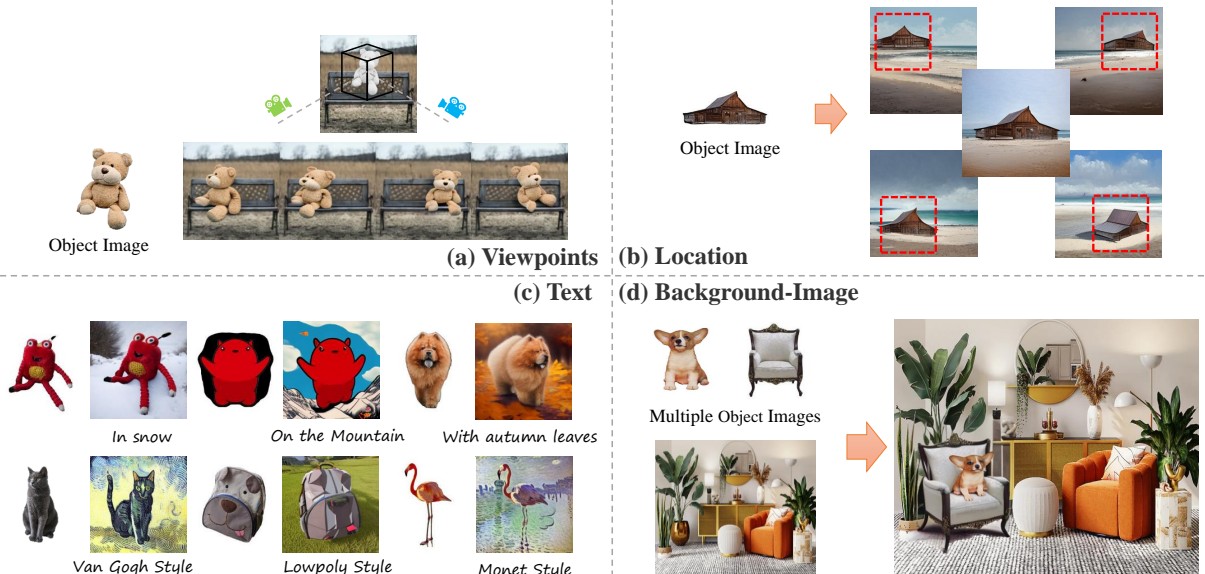

**Figure 1: We propose CustomNet, a novel unified customization method that can generate harmonious customized images without test-time optimization. CustomNet supports explicit (a) viewpoint, (b) location, (c) text, and (d) background image controls while ensuring object identity preservation.**

## ABSTRACT

Incorporating a customized object into image generation presents an attractive feature in text-to-image (T2I) generation. Some methods finetune T2I models for each object individually at test-time, which tend to be overfitted and time-consuming. Others train an extra encoder to extract object visual information for customization efficiently but struggle to preserve the object's identity. To address these limitations, we present CustomNet, a unified encoder-based object customization framework that explicitly incorporates 3D novel view synthesis capabilities into the customization process. This integration facilitates the adjustment of spatial positions and viewpoints, producing diverse outputs while effectively preserving the object's identity. To train our model effectively, we propose a dataset construction pipeline to better handle real-world objects and complex backgrounds. Additionally, we introduce delicate designs that enable location control and flexible background control through textual descriptions or user-defined backgrounds. Our method allows for object customization without the need of test-time optimization, providing simultaneous control over viewpoints, location, and text. Experimental results show that our method outperforms other customization methods regarding identity preservation, diversity, and harmony.

## CCS CONCEPTS

• **Computing methodologies → Computer vision**.

## KEYWORDS

Diffusion models, Object Customization

## 1 INTRODUCTION

Recently, diffusion-based models have achieved new state-of-the-art in text-to-image (T2I) generation [25, 28, 29, 31], allowing ordinary users to synthesize text-complied images. Besides, additional control conditions like layout, style, and depth are applied to these T2I diffusion models [18, 24, 45], achieving fine-grained controls on the synthesized images. Customization, another control dimension in T2I diffusion models, has received significant attention. It allows

users to incorporate objects from reference images into the generation while preserving the object identity. Pioneering works such as Dreambooth [30] and Textual Inversion [9] use a few images of the same object to finetune the parameters of diffusion models or learn concept textual embeddings through an iterative optimization process. Although these optimization-based techniques excel at maintaining object identity, they suffer from certain drawbacks, such as time-consuming optimization for each object and a tendency to be overfitted when only a single image is provided.

Consequently, researchers have started exploring encoder-based methods [16, 18, 22, 40, 42] for efficient customization. These methods train an encoder to encode visual concepts of objects to an embedding. Once trained, users can use the encoder to get the object image embedding and send it into the denoising process during inference, achieving a speed comparable to the standard diffusion model sampling process. However, simply injecting an image into a compressed concept embedding often leads to inadequate identity preservation [16, 42]. Several methods further propose to enhance detail preservation by introducing local features [22, 40], which still cannot preserve complex textures well.

Besides identity-preserving and efficiency, a user may also want to simultaneously change the object viewpoints, which is hard for the customization methods mentioned above. Novel View Synthesis (NVS) diffusion models are capable of addressing viewpoint control. Recent methods [2, 39] regard NVS as an image-to-image translation task and train diffusion models in certain specific categories. Zero-1-to-3 [19], which leverages the massive synthetic 3D object multiple view dataset Objaverse [8] to train a camera pose conditioned diffusion model, extending the generalization capability from a single category to the open world. However, directly applying an NVS diffusion model like Zero-1-to-3 into the object customization task poses several challenges due to its inherent limitations: **1)** It is solely capable of generating centrally positioned objects, lacking the ability to place them in alternative locations in the synthesized image; **2)** It can not generate diverse backgrounds, being restricted to a simplistic white background; **3)** It also lacks the high-level semantic information control like text prompts. These constraints significantly hinder its applicability in object customization tasks.

To efficiently generate harmonious customized images under the control of viewpoints, location, and text while ensuring object identity preservation, We introduce *CustomNet*. CustomNet is a unified encoder-based object customization framework that facilitates diverse viewpoints in text-to-image diffusion models. Unlike previous optimization-based and encoder-based methods that simply rely on text-image paired datasets, CustomNet needs to be trained on more complex data, *i.e.*, a target image with text description and multi-view images of the objects which come from the image. Therefore, we design a dataset construction pipeline that effectively utilizes synthetic multi-view data and massive natural images to better handle real-world objects and complex background relationships for training. Moreover, based on latent diffusion model (LDM) [29] architecture, we design dual cross attention to support both viewpoints and text control for the spatial Transformer in the LDM UNet, and adjust object size and location by concatenating the transformed reference object image with the UNet input. We can further apply style editing to customization through the text-control branch, as the style condition derives from the text prompt

in the data. We also extend CustomNet to reference background image inpainting applications that can receive multiple objects and background images for more flexible customization.

Built upon those designs, CustomNet can achieve harmonious customization with identity preservation and diverse control without test-time optimization, as shown in Fig. 1. We summarize our contributions as follows:

- We propose CustomNet, a unified framework for object customization that explicitly incorporates 3D novel view synthesis capabilities. CustomNet ensures superior preservation of the object's identity, allowing for simultaneous customization of the viewpoint, location of the object, text, and background-image, without test-time optimization.
- We develop a novel dataset construction pipeline that effectively leverages synthetic multi-view data and massive natural images to better customize real-world objects and complex backgrounds more harmoniously.
- Experimental results demonstrate that the proposed CustomNet outperforms existing customization methods regarding identity preservation, diversity, and harmony of the customized results.

## 2 RELATED WORK

**Object customization with text-to-image diffusion models**. With the promising progress of text-to-image diffusion models [11, 25, 28, 29, 31, 33, 34], researches explore to capture the information of a reference object image and maintain its identity throughout the diffusion model generation process, *i.e.*, object customization. These methods can be broadly classified into optimization-based techniques and encoder-based approaches. Optimization-based methods [5, 9, 20, 30] can achieve high-fidelity identity preservation; however, they are time-consuming and may sometimes result in overfitting. In contrast, current encoder-based methods [16, 18, 35, 42] enable zero-shot performance but may either lose the identity. To address this issue, several methods ELITE [40], Subject-Diffusion [22] have been proposed to enhance detail preservation by introducing local features, which still only generate images that resemble the reference images in content and style and do not allow for versatile viewpoint control. This limitation makes it difficult to achieve harmonious results. In contrast, our proposed Custom-Net aims to preserve high fidelity while supporting controllable viewpoint variations, thereby achieving more diverse outcomes.

**Image harmonization**. In image composition, a foreground object is typically integrated into a given background image to achieve harmonized results. Various image harmonization methods [3, 7, 10, 37] have been proposed to further refine the foreground region, ensuring more plausible lighting and color adjustments [4, 6, 41]. However, these methods focus on low-level modifications and cannot alter the viewpoint or pose of the foreground objects. In contrast, our proposed CustomNet achieves flexible background generation using user-provided images and offers additional viewpoint control and enhanced harmonization.

**3D novel view synthesis** aims to infer the appearance of a scene of novel viewpoints based on one or a set of images of a given 3D scene. Previous methods have typically relied on classical techniques such

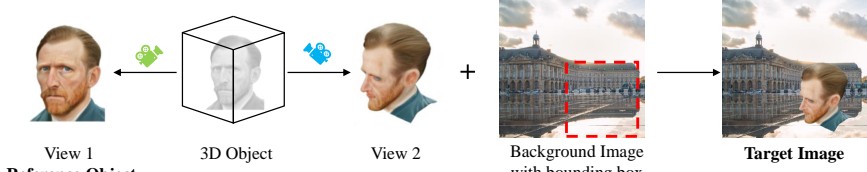

(a) Pipeline 1: data construction from 3D object.

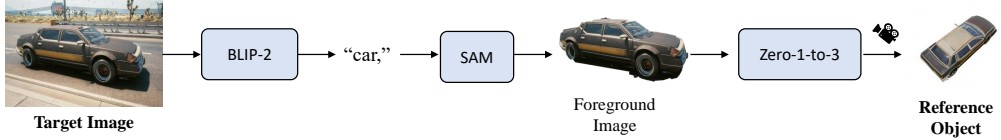

(b) Pipeline 2: data construction from single image.

**Figure 2: Dataset construction pipeline from (a) 3D objects and (b) single image .**

as interpolation or disparity estimation [26, 46], as well as generative models [1, 36]. More recently, approaches based on Scene Representation Networks (SRN) [32] and Neural Radiance Fields (NeRF) [13, 23, 44] have been explored. Furthermore, diffusion models have been introduced into novel view synthesis [19, 39]. Zero-1-to-3 [19] propose a viewpoint-conditioned diffusion model trained on large synthetic datasets, achieving excellent performance in single-view 3D reconstruction and novel view synthesis tasks. Our CustomNet incorporates the 3D capabilities to diffusion models for object customization, generating outputs with diverse viewpoints while preserving the identity.

## 3 METHOD

Given a reference background-free [1] object image $x \in \mathbb{R}^{H \times W \times 3}$ with height $H$ and width $W$, we aim to generate a customized image $\hat{x}$ where the object of the same identity can be seamlessly placed in a desired environment (*i.e.*, background) harmoniously with appropriate viewpoint and location variations. As illustrated in Fig. 3, we propose CustomNet, a novel unified architecture designed to achieve this given object customization conditioned on the viewpoint $R$ (where $R$ represents a vector that contains the relative camera polar angle $\theta$ and azimuth angle $\phi$ of the object in desired viewpoints), object location $L$ (bounding box), and text prompt $T$ that describes the generated image:

$$\hat{x} = \text{CUSTOMNET}(x, R, L, T). \quad (1)$$

In the following, we first introduce the dataset construction pipeline that helps to obtain such paired data: $\{\hat{x}, x, R, L, T\}$, which facilitates harmonious customization with both 3D synthetic dataset and natural image dataset. Then based on LDM [29], we further design the dual cross-attention and Unet input concatenation that makes our model support viewpoints, location, and text control simultaneously. Moreover, similar to other inpainting methods [18, 42], CustomNet can also be extended to the inpainting-like

---

[1]Background-free images can be easily obtained by segmentation methods, *e.g.*, SAM [14].

customization that supports giving the reference background image instead of text prompts.

### 3.1 Dataset Construction Pipeline

As shown in Fig. 2 (a), we can acquire multi-view object images and their corresponding camera view parameters from existing 3D datasets, such as Objaverse [8]. However, these datasets only include object images without backgrounds (usually with a pure white background), and the objects are rendered only at the center of the images. This is not suitable for customization tasks. As a naive solution, we can first collect another background image and specify a random bounding box $L$ located in the background image, then we resize the object and perform mask-blending with the object image and background images in the bounding box. The blended image is the target image $\hat{x}$ and we use another view of the object and relative camera parameters from the 3D datasets as $x$ and $R$, respectively. We use BLIP-2 [17] to caption the textual descriptions of the blended images for the text prompts $T$.

However, since the composition between the object and background would be unreasonable (*i.e.*, the object is placed into the background disharmoniously) and the blended target image is unrealistic, the model trained on them often generates a disharmonious customized image, *e.g.*, the objects float over the background (see Sec. 4.3).

To alleviate this problem, we propose a dataset construction pipeline that is the reverse of the above-mentioned way, *i.e.*, directly utilizing natural images as the target image and extracting objects from the image as the reference. The specific pipeline is shown in Fig. 2 (b). For a natural image $\hat{x}$, we first use BLIP-2 to extract the foreground object with the instruction {"image": image, "prompt": "Question: What foreground objects are in the image? find them and separate them using commas. Answer:"}. Then we feed the object and its corresponding text to SAM. SAM receives text as input and outputs both the bounding box $L$ and the segmentation mask of the corresponding object. We use a bounding box and a segmentation mask to crop the object and synthesize a novel view of the object as $x$ by Zero-1-to-3 with randomly sampled relative viewpoints $R$.

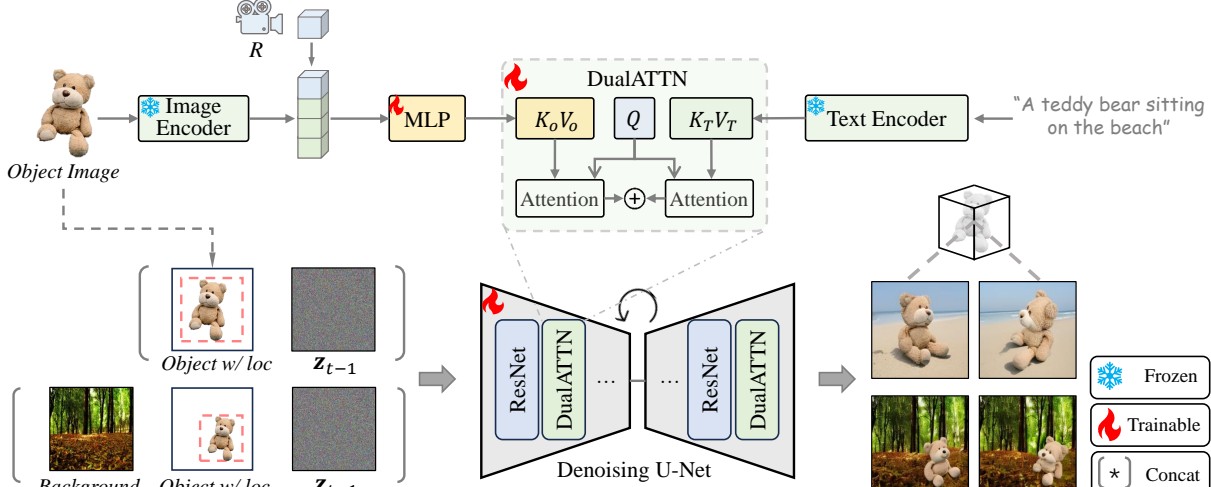

**Figure 3: Overview of CustomNet. CustomNet can simultaneously control viewpoint, location, and text in a unified framework. In bottom-left, The reference object of desired size and location is concatenated with the UNet input. In top-left, the viewpoints embedding is concatenated with object image embedding, which is sent into an MLP module. In top-right, the text prompt embedding is sent to the Dual Attention module to control the generation with the object viewpoints.**

The textual description $T$ of the image can be also obtained using the BLIP-2 model. In this way, we can synthesize a large amount of data pairs from natural image datasets, like OpenImages [15].

## 3.2 Object Viewpoint Control

To enable synthesizing a target customized image complied with the given viewpoint parameter $R$ ($[\theta, sin(\phi), cos(\phi)]$), we follow the view-conditioned diffusion method introduced by Zero-1-to-3. As shown in Fig. 3 (the left-top part), we first apply a pre-trained CLIP [27] image encoder to encode the reference background-free object image into an object embedding, containing high-level semantic information of the input object. Then the object embedding is concatenated with $R$ and passed through a trainable lightweight multi-layer perception (MLP). The fused object embedding further passes to the denoising UNet as a condition with the cross-attention mechanism to control viewpoints of the synthesized images. Even with multi-view datasets, it is worth noting that the explicit viewpoints control is of vital importance to perform object variation and ensure identity consistency, as discussed in 4.3.

## 3.3 Object Location Control

We further control the object location in the synthesized image by concatenating the reference object image with the desired location and size to the UNet input $z_{t-1}$, where $z_{t-1}$ represents the noisy latent at the time step $t-1$. The process is illustrated in Fig. 3 (the left-bottom part). The desired location and size $L$, represented as a bounding box $[x, y, w, h]$, is the object's desirable location in the target image. Then, we resize the reference object image into the size of $[w, h]$ and place its left-top corner at the $[x, y]$ coordinate of a background-free image (this image is the same size as the target image being denoised but without background), which is represented as $x_L$. The additional concatenated reference object

image helps the model synthesize the desired image while keeping the identity and the texture details [19, 29]. Note that Zero-1-to-3 directly concatenates the centrally-located reference object to the UNet input, which can only synthesize an image where the object is centered. Our method enables synthesizing the target object at the desired position with the proposed explicit location control.

## 3.4 Flexible Background Control

We first introduce how to incorporate text prompts to control the background generation, then extend it to inpainting-like customization with the reference background image.

**Text prompt Control.** CustomNet is required to generate an appropriate background based on the textual description $T$. Different from Zero-1-to-3, which solely accepts the object embedding without textual descriptions for background, we propose a dual cross-attention conditioning strategy that accepts both the fused object embedding with viewpoint control and textual descriptions for background. The dual cross-attention mechanism integrates the fused object embedding and the textual embedding through two distinct cross-attention modules. Specifically, we first employ the CLIP text encoder to obtain the textual embeddings and subsequently inject them into the denoising UNet, along with the fused object embedding, using the DUALATTN:

$$\text{DUALATTN}(Q, K_o, V_o, K_T, V_T)$$
$$= \text{Softmax}(\frac{QK_o^T}{\sqrt{d}})V_o + \text{Softmax}(\frac{QK_T^T}{\sqrt{d}})V_T, \quad (2)$$

where the query features $Q$ come from the UNet, while $K_o, V_o$ are the object features projected from fused object embeddings with viewpoint control, and $K_T, V_T$ are the textural embeddings, $d$ is the dimension of the aforementioned feature embeddings. This

straightforward yet effective design enables us to achieve accurate background control without affecting the object viewpoint control. **Background Image Inpainting.** In many practical scenarios, users desire to seamlessly insert objects into pre-existing background images with specific viewpoints and locations. To achieve this, we extend the input channels of the UNet by concatenating the provided background image channel-wise, following the Stable Diffusion inpainting pipeline. Consequently, the diffusion model accepts $[z_{t-1}, object\ w/\ loc, x_{bg}]$ as inputs. Note that in this mode, the textual description is optional, allowing for straightforward input of NULL to the text prompt. In comparison to existing image composition methods [42] which often struggle with identity loss issues, our method offers viewpoint and location control over objects and enhanced identity preservation in 4.2.

## 3.5 Training Strategies

**Model Training.** Given paired images (object image and target image) with corresponding relative camera viewpoint, resized object image in specified locations, and text prompt $\{x, x_{tgt}, R, x_L, T\}$, we train our diffusion model conditioned on these explicit controls. We adopt the latent diffusion model (LDM) [29] architecture, which contains a variational auto-encoder (VAE) with an encoder $\mathcal{E}$, decoder $\mathcal{D}$, and an UNet denoiser $\epsilon_\theta$. LDM uses VAE to convert the image from pixel-level space to latent space, and performs the diffusion-denoising process in the latent space rather than the pixel space for efficiency. The optimization objective is:

$$\min_\theta \mathbb{E}_{z_t,\epsilon,t} \parallel \epsilon - \epsilon_\theta(z_t, x, R, x_L, T) \parallel^2, \tag{3}$$

Once the denoising UNet $\epsilon_\theta$ is trained, we can perform harmonious customization conditioned on the target viewpoint, location, and text prompt with CustomNet.

## 3.6 Classifier-free Guidance

Classifier-free diffusion guidance [12] is a method for adjusting the quality and diversity of diffusion generations. In CustomNet, we divide the conditions into two parts: image ($x, R, x_L$, which is related to object viewpoint and location) and text ($T$ for textual description). For sampling , we set two guidance scales ($S_I, S_T$) to control their influence respectively as follows:

$$\hat{\epsilon}_\theta(z_t, x, R, x_L, T) = \epsilon_\theta(z_t, \emptyset_x, \emptyset_R, \emptyset_{x_L}, \emptyset_T)$$
$$+ S_I \cdot (\epsilon_\theta(z_t, x, R, x_L, \emptyset_T) - \epsilon_\theta(z_t, \emptyset_x, \emptyset_R, \emptyset_{x_L}, \emptyset_T)) \tag{4}$$
$$+ S_T \cdot (\epsilon_\theta(z_t, x, R, x_L, T) - \epsilon_\theta(z_t, x, R, x_L, \emptyset_T))$$

where $\emptyset_*$ is set the $*$ condition to null. During training, we randomly drop the image part by 5% and the text part by 15%.

## 4 EXPERIMENTS

## 4.1 Training Datasets and Implementation Details

As introduced in Sec. 3.5, We use multi-view synthetic dataset Objaverse [8] and additionally obtained background images from the web to synthesize blended images. Then We use natural image dataset OpenImages-V6 [15] filtered as BLIP-Diffusion [16] to construct data pairs with the pipeline. A total of (250+500)K data pairs are constructed for model training with sampling ratio 5% : 95%,

respectivaley. We exploit the Zero-1-to-3 checkpoint as the model weight initialization. For training, we employ AdamW [21] optimizer with a constant learning rate $2\times10^{-5}$ for 500K optimization steps. The total batch size is 96, and about 6 days are taken to finish the training on 8 NVIDIA-V100 GPUs with 32GB VRAM.

## 4.2 Comparison to Existing Methods

We compare our CustomNet to the optimization-based methods Textual Inversion [9], Dreambooth [30], and encoder-based method GLIGEN [18], ELITE [40], BLIP-Diffusion [16], IP-Adapter [43]. We use their official implementation (for GLIDEN, ELITE, and BLIP-Diffusion, IP-Adapter) or the diffusers implementations [38] (for Textual Inversion, Dreambooth) to obtain the results. Note that Dreambooth requires several images of the same object to finetune.

Figure 4 shows the images generated with different methods. We see that the methods GLIGEN, ELITE, BLIP-Diffusion, IP-Adapter and the optimization-based method Textual Inversion are far from the identity consistent with the reference object. Dreambooth and the proposed CustomNet achieve highly promising harmonious customization results, while our method allows the user to control the object viewpoint easily and obtain diverse results. In addition, our method does not require time-consuming model fine-tuning and textual embedding optimization.

We also evaluate the synthesized results quantitatively. All methods apply 26 different prompts to perform customizations 3 times randomly on 50 objects, a total of above 3700 cases. Following [16], we calculate the visual similarity with the CLIP image encoder and DINO encoder, denoted as **CLIP-I** and **DINO-I**, respectively. We measure the text-image similarity with CLIP directly, denoting **CLIP-T**. Tab. 1 shows the quantitative results, where CustomNet achieves better identity preservation (**DINO-I** and **CLIP-I** than other methods. Meanwhile, CustomNet shows comparable capacity to the state-of-the-art methods regarding textual control (**CLIP-T**). We also conducted a **user study** and collected 2700 answers for *Identity similarity* (**ID**), *View variation* (**View**), and *Text alignment* (**Text**), respectively. As shown in the right part of Tab. 1, most participants prefer CustomNet in all three aspects (76.11%, 56.67%, 64.67%). For the test-time-consuming comparison, CustomNet and other encoder-based methods are faster than optimization-based methods in general.

**Comparison to Inpainting-based Methods** Existing inpainting-based methods (SD-Inpainting model, Paint-by-Example [42], GLIGEN [18]) can also place a reference object in the desired background in an inpainting pipeline. Given an object, the background can be inpainted with textual descriptions in the SD-Inpainting model, while this kind of method easily suffers from unreal and disharmonious results and cannot cast variations to the reference object. Our CustomNet can obtain more harmonious customization with diverse viewpoint control. Another line of methods Paint-by-Example and GLIGEN can inpaint the reference object to a given background image. From Fig. 5, we see that they cannot maintain the identity and differ significantly from the reference object.

**Comparison to Zero-1-to-3** We show our improvements of Zero-1-to-3 and its limitations in Fig. 6. In the 1st, and 2nd row, we compare with Zero-1-to-3 the ability to control the object location generation. As Zero-1-to-3 can only control polar angle $\theta$, azimuth

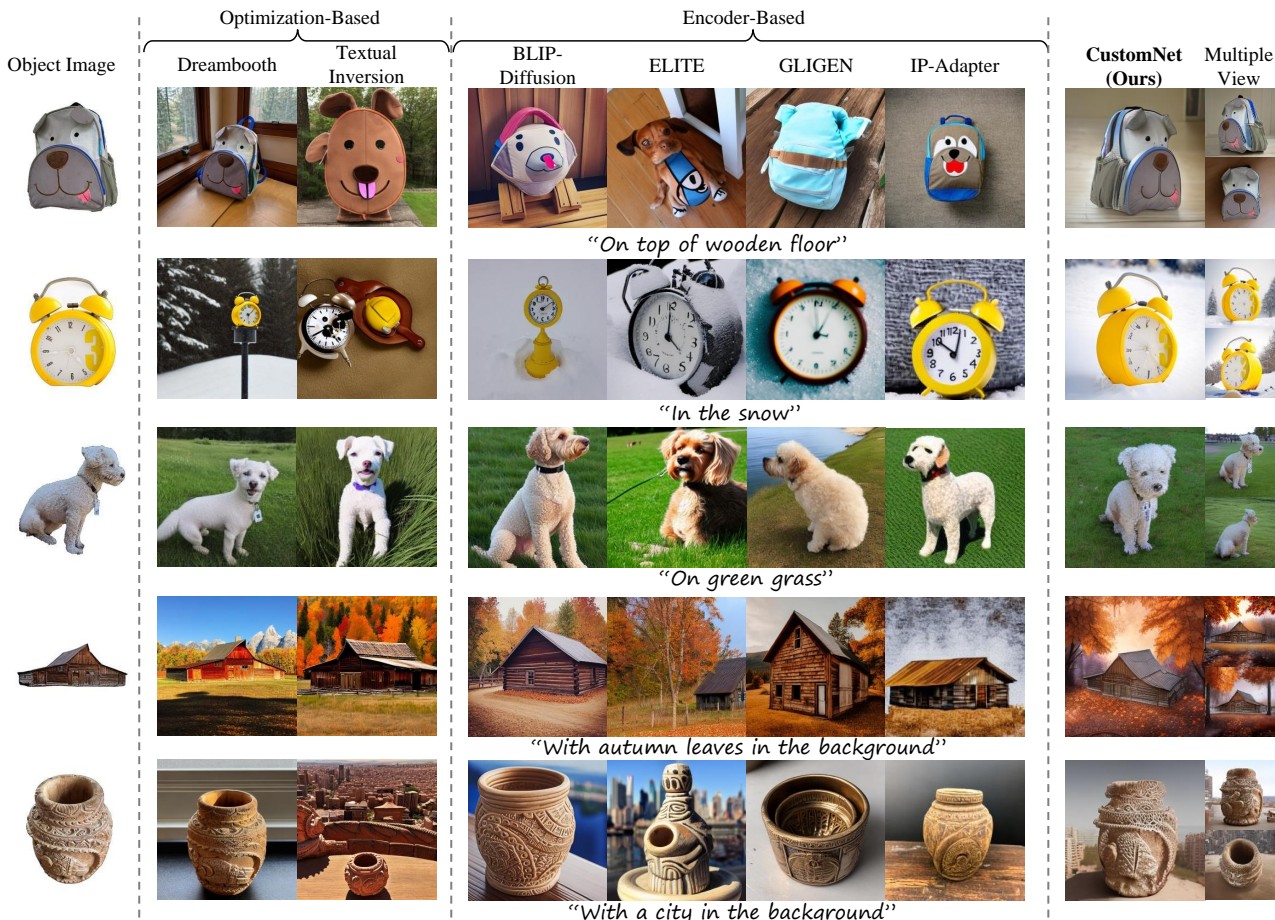

**Figure 4: Qualitative comparison. Our CustomNet demonstrates superior capacities in terms of identity preservation, viewpoint control, and harmony of the customized image.**

**Table 1: Quantitative Comparison. We compute DINO-I, CLIP-I, CLIP-T following [16]. We also conducted a user study to measure subjective metrics: ID, View, Text representing identity preservation, viewpoints variation, and text alignment, respectively. The last column compares the test-time consumption of different methods.**

| Method | DINO-I ↑ | CLIP-I ↑ | CLIP-T ↑ | ID ↑ | View ↑ | Text ↑ | Time (s) ↓ |
|---|---|---|---|---|---|---|---|
| DreamBooth [30] | 0.6333 | 0.8019 | 0.2276 | 0.1322 | 0.0833 | 0.1367 | ∼600 |
| Textual Inversion [9] | 0.5116 | 0.7557 | 0.2088 | 0.0111 | 0.0911 | 0.0433 | ∼1500 |
| BLIP-Diffusion [16] | 0.6079 | 0.7928 | 0.2183 | 0.0511 | 0.0833 | 0.0444 | ∼8 |
| ELITE [40] | 0.5101 | 0.7675 | **0.2310** | 0.0078 | 0.0656 | 0.1033 | ∼5 |
| GLIGEN [18] | 0.5587 | 0.8152 | 0.1974 | 0.0233 | 0.0678 | 0.0156 | ∼23 |
| IP-Adapter [43] | 0.5801 | 0.8072 | 0.1919 | 0.0133 | 0.0422 | 0.0100 | ∼11 |
| **CustomNet (Ours)** | **0.7742** | **0.8164** | 0.2258 | **0.7611** | **0.5667** | **0.6467** | ∼8 |

angle $\phi$, and radius $r$ (the camera distance to the object) of the central object, users can not control the object location in their settings directly, so we apply our location control method on Zero-1-to-3. We resize the object and move it to the location in different bounding boxes (top left and top right in 1st row, bottom left and bottom right in 2nd row.). We can see that Zero-1-to-3 generates

distortion in the generated images. This is because Zero-1-to-3 is only trained on the object-centered image, it can not really control the object location.

In the 3rd, and 4th row, We compare with Zero-1-to-3 the ability of novel-view synthesis. We can see from the figure that Zero-1-to-3 fails to generate reasonable geometry for the dog and cartoon

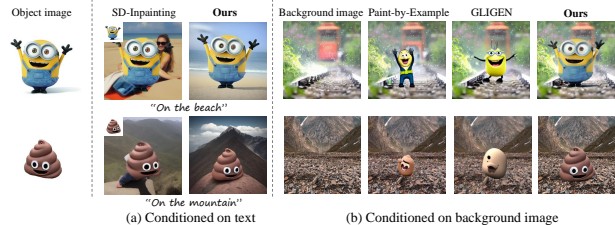

(a) Conditioned on text     (b) Conditioned on background image

**Figure 5: Comparison to existing textual background inpainting method SD-Inpainting model and foreground object inpainting model Paint-by-Example and GLIGEN. Our CustomNet can achieve a more harmonious output with diverse viewpoint changes while preserving identity.**

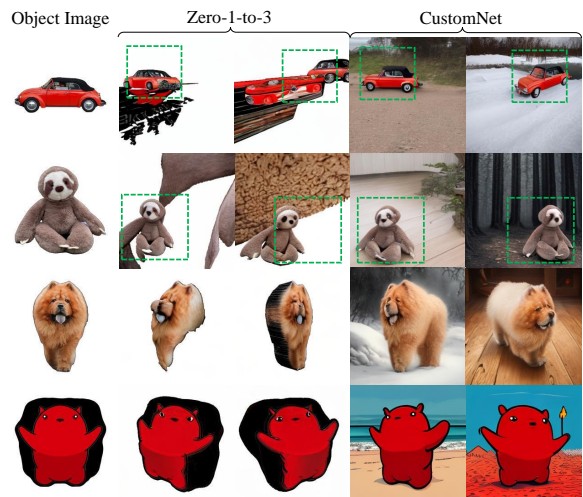

**Figure 6: The comparison of CustomNet and Zero-1-to-3. In the 1st and 2nd rows, CustomNet can control the object location well. In the 3rd and 4th rows, CustomNet synthesizes a better novel view with the help of real-world dataset. Besides, CustomNet generates harmonious images with the control of different text prompts.**

## 4.3 Ablation Studies and Analysis

We conduct detailed ablation studies to demonstrate the effectiveness of each design in CustomNet and the necessity of explicit viewpoints control for identity preservation in harmonious customization.

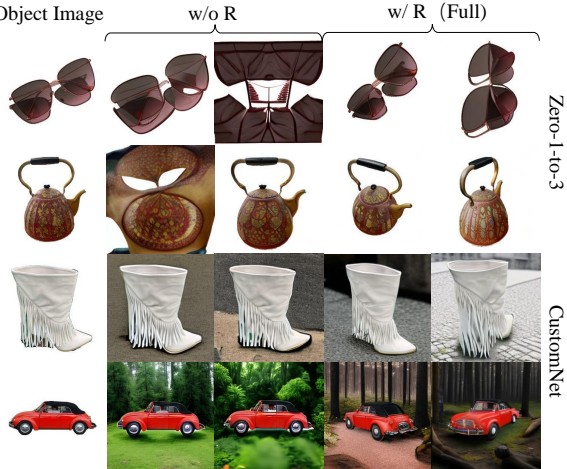

**Figure 7: Explicit viewpoints control. Without the explicit viewpoint parameters *R*, Zero-1-to-3 tends to generate images that cannot change the viewpoint or have undesired artifacts; CustomNet easily obtains copy-pasting effects without *R*, even though it is trained on the multi-view dataset.**

character. Our CustomNet, training with our real-world constructed datasets, has a better comprehension of the object geometry. We can generate a normal dog and a reasonable cartoon character in multi-views.

Moreover, CustomNet supports diverse text prompt condition generation that Zero-1-to-3 can not, which is more practical for customization. In all rows, our CustomNet generates harmonious customized images with the control of different text prompts.

***Explicit viewpoint control is the key for customization that enables simultaneous viewpoint alteration and object identity preservation.*** We conduct a comparison in terms of with and without explicit viewpoint control parameters $R$ on the original Zero-1-to-3 model. As shown in the left part of Fig. 7, models trained without viewpoint conditions tend to generate images that cannot change the viewpoint or have undesired artifacts. This is the same on CustomNet. Specifically, as shown in the right part of Fig. 7, without the explicit camera pose control, our model can only obtain copying-and-pasting effects, even though it is trained with the multi-view dataset. Note that in this setting, we also concatenate the object image into the UNet input, otherwise, it cannot preserve adequate identity. This phenomenon may indicate that even with multi-view data to train, an explicit additional view control is still necessary, which allows the model to better distinguish between the distribution of images with different views, rather than learning a mixed or average of different views.

***Pretraining with massive multi-view data is important to preserve identity in CustomNet.*** We adopt Zero-1-to-3 as the initialization, *i.e.*, our CustomNet is pre-trained with massive multi-view Objarverse data, so that view-consistency information has been already encoded into the model. When we train CustomNet from the SD checkpoint (see the 2nd column in Fig. 8), the synthesized images cannot maintain view consistency from other viewpoints and suffer from quality degradation.

***Input concatenation helps better maintain texture details.*** Previous methods [22, 40] also try to integrate local features to maintain texture details in synthesized images. In our method, concatenating the reference object to the UNet input can also preserve textures. Without the concatenation (see the 3rd column in Fig. 8), the color, shape, and texture of the generated images differ significantly from the reference object image. We also note that the model

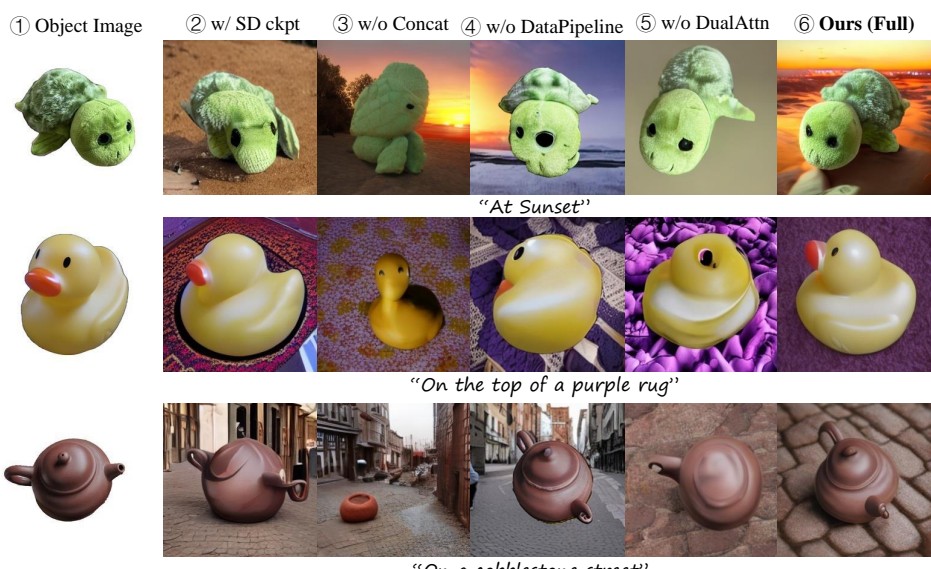

① Object Image    ② w/ SD ckpt    ③ w/o Concat    ④ w/o DataPipeline    ⑤ w/o DualAttn    ⑥ **Ours (Full)**

*"At Sunset"*

*"On the top of a purple rug"*

*"On a cobblestone street"*

**Figure 8: Ablation Study. *w/ SD ckpt*: initialize model weights with Stable-Diffusion pretrained checkpoints. *w/o Concat*: do not concatenate object with UNet input. *w/o DataPipeline*: do not use the dataset construction pipeline for OpenImages. *w/o DualAttn*: concatenate image and text embedding together and use shared cross-attention modules. *Ours(Full)*: our full model presents harmonious results with diverse controls. All images in each row are generated with the same viewpoints and text.**

would generate copying-and-pasting images without any view variations when we do not adopt explicit viewpoint control (see Fig. 7). This is to say, the combination of input concatenation and explicit view conditions enables precise and harmonious customization.

***Our data construction pipeline enables more harmonious outputs.*** We adopted a new data construction pipeline for utilizing the OpenImages dataset in Sec. 3.5. Without this design, the model trained with only the data constructed by the naive combination between multi-view object images in Objaverse and background images can result in unrealistic and unnatural customized results, usually leading to artifacts that object is 'floating' on the background (see the 4th column in Fig. 7).

***Dual cross-attention enables disentangled object and text controls.*** We introduce dual attention for the disentangled object-level control and text control. When directly concatenating the text embedding and fused object embedding as the condition to be injected into the attention layers of UNet, the model tends to learn a coupled control with viewpoint parameters and textual description. As a result, the viewpoint control capacity would degrade significantly and the model cannot generate the desired background (see 5th column in Fig. 8).

***Classifier-free guidance weights effects.*** CustomNet is controlled by both image-related and text conditions for generation using classifier-free guidance. In Tab. 2, we compare different classifier-free guidance weights effects and empirically choose $S_I = 3, S_T = 7.5$ for comparison experiments.

## 5    CONCLUSION

We present CustomNet, a novel unified encoder-based diffusion object customization approach that explicitly incorporates 3D novel

**Table 2: Classifier-free guidance weights effects. We compute DINO-I, CLIP-I CLIP-T following [16]to compare different classifier-free guidance weights effects.**

| Method | DINO-I ↑ | CLIP-I ↑ | CLIP-T ↑ |
|---|---|---|---|
| $S_I = 1.5, S_T = 5$ | 0.7293 | 0.7643 | 0.2144 |
| $S_I = 1.5, S_T = 7.5$ | 0.7224 | 0.7604 | 0.2230 |
| $S_I = 3, S_T = 5$ | 0.7716 | **0.8185** | 0.2119 |
| $S_I = 3, S_T = 7.5$ | **0.7742** | 0.8164 | **0.2258** |

view synthesis for enhanced identity preservation and viewpoint control. We develop a dataset construction pipeline to handle real-world objects and complex backgrounds effectively. Moreover, we introduce extra necessary designs for location control and flexible background control through textual descriptions or provided background images, which improve the novel view synthesis model (*e.g.*Zero-1-to-3), and show potential for bridging the gap between 3D novel-view synthesis and text-to-image object customization. Our experiments show that CustomNet enables identity preserving and diverse object customization while controlling location, viewpoints, and background simultaneously.

## 6    LIMITATION AND FUTURE WORKS

CustomNet may be limited to some complex prompts and rarely seen objects and scenery, and existing diffusion models trained with web-scale datasets would also encounter these problems. Current research shows that fine-tuning the model with specific small but high-quality datasets may make the generation better. We will explore it in future work.

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
