# OpenReview forum: "CustomNet: Object Customization with Variable-Viewpoints in Text-to-Image Diffusion Models"
_acmmm.org/ACMMM/2024/Conference — MM2024 Poster_

### Official Review · Reviewer_5AdS · 2024-05-24

**Rating:** 4
**Confidence:** 3

**Summary:**

The paper introduces a new unified encoder framework named CustomNet for object customization in text-to-image (T2I) diffusion models, incorporating 3D novel viewpoint synthesis capabilities. It designs a well-suited dataset augmentation construction process and effectively integrates multiple image generation control conditions into the model using the built dataset, achieving the final output that can utilize various viewpoints, texts, and specified background images.

**Strengths:**

1. The dataset construction process is intuitive and addresses shortcomings in previous approaches, resulting in an ideal dataset for the current task that includes multiple viewpoints and contains text and background data for image synthesis.
2. The proposed method effectively merges multiple control conditions, with each condition's usage and integration process being very intuitive. Experiments also demonstrate its superior performance and efficiency compared to other methods.
3. The paper is well-written with straightforward method flowcharts, allowing for a clear understanding of the ideas and model pipeline. The visualization of experimental results is impressive, effectively showcasing the method's capabilities.

**Limitations:**

1. It appears that the ability of CustomNet to control viewpoints stems from training with multi-view images. However, it seems that there is no direct control over viewpoints during testing. Is it possible to explicitly control the object's angle through text?

2. Why is your inference speed faster than IP-Adapter? There doesn't seem to be any difference in your structures.

3. Although CustomNet can change the target's perspective, the target is still static and it seems that the corresponding action cannot be changed. Is it possible to use some avater method to get image generation that keeps the ID instead of just the static target itself?

**Suitability:**

2

---

### Official Review · Reviewer_CS13 · 2024-06-04

**Rating:** 4
**Confidence:** 2

**Summary:**

This paper presents a unified encoder-based diffusion object customization approach that explicitly integrates 3D novel view synthesis to improve identity preservation and viewpoint control.

**Strengths:**

1. This work employs SAM and Zero-1-to-3 to construct image datasets from various viewpoints.
2. Introducing viewpoint information allows for controlling the target's perspective.
3. The experimental results in Table 1 appear to be highly effective.

**Limitations:**

1. The paper lacks quantitative baseline and ablation experiment analyses. Relying solely on visual results for analysis lacks persuasive strength.
2. Despite not requiring fine-tuning of the entire pre-trained model, this method still demands a substantial amount of time for fine-tuning, i.e., 6 days on 8 V100 GPUs. Additionally, why CustomNet's test-time in Table 1 is faster compared to other methods remains unclear. It seems that the introduction of more complex conditional control information and dual attention mechanisms during the CustomNet training process may contribute to the speed improvement.
3. The authors' approach relies on pre-training weights from Zero-1-to-3, thus necessitating a quantitative comparison with Zero-1-to-3.
4. Were the experiments in Table 1 conducted under identical settings? Additional experimental details are required, including the base model, classifier-free guidance scale, and training data.

**Suitability:**

2

---

### Official Review · Reviewer_XmRC · 2024-06-06

**Rating:** 4
**Confidence:** 4

**Summary:**

This paper proposes an encoder-based object customization framework that enables view and location control without test-time optimization. They also propose a dataset construction pipeline to handle complex cases. The method shows a good ID preservation ability as well as controllability.

**Strengths:**

The method is efficient since it is encoder-based without need of test-time optimization.
The qualitative and quantitative results show the advantages of the method over others.

**Limitations:**

I do have some questions regarding the method details.

1. About Dataset construction pipelines shown in Figure 2. It is said that Pipeline 1 generates disharmonious customized images, so the second pipeline is adopted to alleviate the problem. So my understanding is that these two pipelines are not complementary, instead the latter is an improved version over the former right? If so, how does the second pipeline solve the issue of disharmony? Using the synthesized object from Zero-1-to-3, we still have to put the object back to a background image. So how is the new pipeline better?

2. Line 327 says "We use BLIP-2 to caption the blended images for T". I am wondering how well BLIP-2 works on those disharmonious blended images? For example, the Target image in Figure 2(a), the head blended in a background image, because it is very unusual image.

3. In Figure 3, for UNet, there are two types of input, 1) concatenation of obj and z_{t-1} 2) concatenation of bd, obj and z_{t-1}. In order to feed these two types of input, do we need to adapt input/output channels of the UNet? (change the original 4 channels to 8 and 12?) If we do have to adapt the input/output channels, how does the method deal with these two different types with different channels (8 and 12)?

4. In Figure 4, compared with other encoder-based methods like BLIP-Diffusion, ELITE and IP-Adapter, the proposed method shows way better ID-preservation abilities. But as shown in Figure 8, we see the ID is preserved badly without concatenation of the object image. So I am wondering if the strong ID-preservation ability is simply from concatenation and whether it is a fair comparison over other methods. The concatenation technically leaks the object appearance information as test-time whereas the other encoder-based methods only use the high-level features from the image encoder. In other word, how well do other methods perform with concatenation of the object as well?

To conclude, I think the main contribution is adding view and location control signals to the customization network. But I think the comparison is not very fair to truly show the model performance as stated in point 4. I also think the writing could be improved.

**Suitability:**

2

---

### Meta-Review · Area_Chair_bFH2 · 2024-07-02

**Recommendation:** Accept (Poster)
**Confidence:** 4

**Metareview:**

1. This paper presents a unified encoder-based diffusion object customization approach that explicitly integrates 3D novel view synthesis to enhance identity preservation and viewpoint control.
It also introduces a well-designed dataset augmentation process and effectively incorporates multiple image generation control conditions using the constructed dataset, addressing shortcomings in previous approaches.
The proposed method seamlessly merges multiple control conditions, with each condition's usage and integration being very intuitive.
Experiments demonstrate its superior performance and efficiency compared to other methods.
The paper is well-written with straightforward method flowcharts, providing a clear understanding of the approach.

2. However, there are some weaknesses in this paper. It lacks quantitative baseline and ablation experiment analyses, relying solely on visual results, which reduces its persuasive strength.
Additionally, it does not provide an analysis of why the proposed method is faster than other methods or address its limitations in pose or action editing.

Based on these reasons and all reviews rate borderline accept, I am inclined to accept this paper.